# Influence of Strain Rates during Severe Plastic Strain Processes on Microstructural and Mechanical Evolution in Pure Zinc

**DOI:** 10.3390/ma15144892

**Published:** 2022-07-14

**Authors:** Mariusz Kulczyk, Jacek Skiba, Monika Skorupska, Sylwia Przybysz, Julita Smalc-Koziorowska

**Affiliations:** Institute of High Pressure Physics, Polish Academy of Sciences (Unipress), Sokołowska 29/37, 01-142 Warsaw, Poland; skiba@unipress.waw.pl (J.S.); monikaw@unipress.waw.pl (M.S.); sylwia@unipress.waw.pl (S.P.); julita@unipress.waw.pl (J.S.-K.)

**Keywords:** hydrostatic extrusion, equal channel angular pressing, strain rate, zinc

## Abstract

The study presents an analysis of the influence of the plastic strain rate on the mechanical and structural properties of pure zinc. Thanks to the use of unconventional methods of plastic processing, the process of the equal channel angular pressing (ECAP) and the process of hydrostatic extrusion (HE), the tests were performed in a wide range of plastic strain rates, between 0.04 s^−1^ and 170 s^−1^. Plastic strain rate changes were carried out in the course of the significant plastic strain processes, and not on previously deformed samples. All tests were carried out at a constant value of plastic strain rate, ε ~ 2. A strong influence of the plastic strain rate on changes in the microstructure in zinc was observed during the tests. For the rates in the range of 0.04 s^−1^ to 0.53 s^−1^ its bimodal nature was observed, and in the range of 7 s^−1^ to 170 s^−1^ high homogeneity and evenness of grains related to the processes of continuous dynamic recrystallization was noticed. The effect of the strong homogenization of the microstructure was the increase in mechanical properties, yield point and tensile strength to the maximum values of UTS = 194 MPa, YS = 145 MPa at a strain rate of 170 s^−1^. Compared to the material with a bimodal microstructure, an over seven-fold increase in the elongation value was observed.

## 1. Introduction

Zinc and zinc-based alloys are attractive materials for biomedical applications for biodegradable implants [1,2]. This is due to the excellent biocompatibility and moderate degradation rate in-vivo [3,4]. A big obstacle to their potential use is their low mechanical properties, the yield point of about 20 MPa at an elongation of 12% [5]. Hence, many research works focus on the possibilities of improving the mechanical properties of zinc, both by doping with additives such as Mg, Ca, Sr, Mn or by using unconventional methods that allow microstructure fragmentation, such as the ECAP (equal channel angular pressing) process [5,6,7,8,9]. A major obstacle in the effective disintegrating of the microstructure in zinc is its low recrystallization temperature (−12 °C), which causes dynamic recrystallization processes during plastic strain processes if carried out at room temperature [1]. 

Due to the high susceptibility of these materials to the thermal effects accompanying the processes of high plastic strain, we attempted to investigate the influence of the strain rate on the mechanical and structural properties of zinc. The plastic strain rate, in addition to the degree of plastic deformation, has a significant impact on the efficiency of generating structural defects and thermal effects, thus reducing the effects of strain strengthening. Studies on the influence of the strain rate on the mechanical and structural properties are mainly carried out on previously deformed materials or on coarse-crystalline materials using standard static or dynamic tests. Such studies have been carried out for many materials, mainly after the ECAP process, such as magnesium alloys, aluminum alloys or titanium, but also after ARB (accumulative roll-bonding) or cold rolling processes [10,11,12,13,14]. In the case of zinc, such work was carried out in dynamic compression tests, where the leading role of the continuous dynamic recrystallization process at higher compression rates (~0.5 s^−1^) was observed [1,15].

In the present study, the authors, for the first time, attempted to analyse the influence of the strain rate generated directly during severe plastic deformation processes in a wide range between 0.04 s^−1^ a 170 s^−1^. The strain rate parameter changed during the SPD processes, but not during the mechanical tests on previously strained samples, which is similar to the data reported in the literature on the subject. Such a wide range of strain rates was possible thanks to the use of the ECAP method in the range of lower strain rates of 0.04–0.53 s^−1^ and hydrostatic extrusion (HE) method for a strain rate in the range of 7 s^−1^ up to 170 s^−1^. Although the ECAP method is a method quite widely used for strong deformation of materials, the HE method is a unique technology developed by the authors of the paper. The effectiveness of the use of the hydrostatic extrusion process as a method leading to the fragmentation of the metals and metal alloys’ microstructure was repeatedly presented by the authors in the literature on the subject for a wide range of materials. This applies to very flexible materials, such as aluminum alloys or copper alloys, but also to hard-deformable materials, such as titanium or austenitic steels [16,17,18,19,20,21]. The hydrostatic extrusion process was also used by many authors in zinc alloys with the addition of magnesium where, at a cumulative strain of ε_cum_ ~ 3.55 very high mechanical properties were observed at the level of ultimate tensile strength UTS = 515 MPa and yield strength YS = 375 MPa, significantly exceeding the values reported in the literature [22].

## 2. Materials and Methods

The tested material was zinc with a purity of 99.9%. Due to the heterogeneity of the microstructure in the original state, the material was annealed at 150 °C for 30 min. From bars with a diameter of 30 mm, samples with a square cross-section of 10 mm × 10 mm and a length of 60 mm were cut by machining for the ECAP process. The ECAP process was carried out in two passes using the C method, i.e., with a 180° rotation between successive passes. The ECAP process was carried out in cold conditions in a 90° chamber, at three different rates of plastic strain, i.e., 0.04 s^−1^, 0.13 s^−1^ and 0.53 s^−1^. The rate was controlled by the piston travel speed. The total actual strain in each of the tested samples was ε ~ 2. The samples for the hydrostatic extrusion process were circular, with a diameter of 16 mm. The process of hydrostatic extrusion was carried out in cold conditions with intensive cooling in the zone of plastic strain. The actual strain was the same as in the ECAP process, i.e., ε ~ 2, which corresponded to extrusion to the final diameter of 6 mm. The HE processes were carried out at three different rates of plastic strain, i.e., 7 s^−1^, 50 s^−1^ and 170 s^−1^. The rate was controlled by the volume of the pressure medium and the rate of its compression. The process of hydrostatic extrusion was carried out on dies with an apex angle of 2α = 45°. The specificity of the hydrostatic extrusion process has been widely presented in previous work [23].

Microstructural observations for the initial material and after plastic strain were carried out on the Nikon Eclipse LV150 light microscope LM and the FEI TECNAI G2 F20 transmission electron microscope TEM. The grain sizes before and after the HE and ECAP processes were determined with the image analysis method using Micrometer software (version 1.0, prof. Tomasz Wejrzanowski Warsow Uniwersity of Technology, Faculty of Materials Science and Engineering Warsaw, Poland) [24]. In each case, the data were based on the obtained LM and TEM images. After the imaging, at least 200 grains selected randomly from the population were outlined and the software calculated the equivalent grain diameter d_2_ (defined as the diameter of the circle with the surface area equal to that of the given grain).

The mechanical properties of the samples were examined in a Zwick/Roell Z250 kN machine (Ulm, Germany) using the static tensile test at room temperature, the strain rate of 0.008 s^−1^ on the standard round samples with length to diameter ratio 5:1 machined along the samples axis.

## 3. Microstructure

Figure 1 shows the basic zinc microstructure after a 30 min annealing at 150 °C. After the annealing process, a homogeneous microstructure was obtained in the form of equiaxial recrystallized grains with an average size d_2_ = 15.3 µm. Figure 2 shows the grain size distribution determined based on the obtained microstructure images, showing a typical character for coarse-crystalline, recrystallized materials. The value of the particle size distribution variation coefficient was Cv_d2_ = 0.5.

Figure 3 shows the microstructure of zinc after the plastic strain with strain rates in the range of 0.04 s^−1^ to 0.53 s^−1^ implemented utilising the ECAP process and in the range of 7 s^−1^ to 170 s^−1^ in the HE process. In the range of the lower rates of plastic strain generated during the ECAP process, the observed microstructure shows a bimodal nature (Figure 3a–c). A fraction of smaller grains of size d_2_~ 1 µm to 5 µm were observed, as well as larger grains with an average size in the range between 15 µm and 30 µm. Medium grain sizes d_2_, along with the coefficient of variation of the Cv_d2_ distribution determined based on the images obtained, are presented in Table 1. The values of the coefficient of variation of the grain size distribution Cv_d2_ were calculated as the ratio of the mean value d_2_ to the standard deviation of the population. The selected grain size distributions are shown in Figure 4. At the lowest plastic strain rate of 0.04 s^−1^, two fractions of grains—a smaller and a larger one—can be clearly seen when analysing the grain size distribution presented in Figure 4. Despite the fact that the average grain size drops more than twice to the size d_2_ = 6.1 μm compared to the unprocessed material, a significant increase in the coefficient of variation of the grain size distribution is observed, which proves the significant heterogeneity of the microstructure, which is manifested in this case by its bimodal nature, as shown in Table 1.

Moreover, as the strain rate increases, the value of the coefficient of variation of the grain size distribution decreases, which proves that the microstructure is homogenised. The areas of the finer grains begin to disappear, and the average grain size slightly decreases. The microstructure shows a completely different nature when a higher strain rate in the range of 7 s*^−^*^1^ up to 170 s*^−^*^1^ is utilised. A fully homogeneous microstructure of the equiaxed grains is observed, with much lower values of the coefficient of variation of the grain size distribution within the Cv_d2_ range of 0.36 to 0.43. The images presented in Figure 3d–f show a strong homogeneity of the microstructure, as well as the grain size distribution with almost 80% presence of grains with the size of d_2_ ~ 5 µm. A slight increase in the coefficient of variation of the grain size distribution Cv_d2_ in the range of high rates of strain can be related to the generated in the process of extrusion, as shown in Figure 5. By analysing the influence of the strain rate on the change of the extrusion pressure during the HE process, a clear increase in pressure is observed in the entire range of the strain rates tested. The higher pressure of the HE process generates stronger thermal effects and a higher homologous temperature of the process [18].

Microstructural changes of a similar nature were observed in the zinc subjected to cold compression processes with a degree of compression of 161%, where, after exceeding the strain rate of 0.5 s^−1^ a homogeneous microstructure was observed with an average grain size of d_2_ = 24 μm [15]. These changes were attributed to the process of continuous dynamic recrystallization that occurred at a sufficiently high degree of material compaction and at a sufficiently high rate of strain. In this study, the microstructural analysis performed after large plastic strain processes with real strain at the level ε ~ 2 indicates that the process of continuous dynamic recrystallization takes place when the strain rate of 7 s^−1^ is exceeded. This phenomenon is also confirmed by images of the microstructure obtained by using a TEM microscope, as shown in Figure 6. In zinc, after the straining process with the lowest strain rate of ~0.04 s^−1^, areas of smaller defective grains are observed, as shown in Figure 6a. After the processes generating higher plastic strain rates (higher than 7 s^−1^), clear, equiaxed recrystallized grains are observed, practically free from defects, which was also seen in the deformed zinc at the highest rate of 170 s^−1^ in Figure 6b.

## 4. Mechanical Properties

Figure 7 shows the characteristics of the static tensile test for two extreme strain rates, the lowest rate of 0.04 s*^−^*^1^ and the highest rate of 170 s*^−^*^1^. The curves obtained reflect clear differences in the structure of the material. At the highest strain rate, when a high degree of homogeneity of the microstructure was observed, a sharp increase in both the ultimate tensile strength (UTS) and elongation at the break was also observed. The results obtained in the static tensile test for all tested strain rates are summarised in Table 2.

In the range of a plastic strain rate between 0.04 s*^−^*^1^ and 0.53 s*^−^*^1^ in which the material was subjected to the ECAP process, a significant increase in zinc strength—both in terms of YS yield strength and UTS ultimate tensile strength (by 150% and 300%, respectively)—was observed compared to the unprocessed material. In the range of these strain rates, the strength values are close to each other. However, the elongation value initially does not change (at 0.04 s*^−^*^1^), but at the strain rate at 0.53 s*^−^*^1^, it grows almost threefold to the level of A ~ 6%. This is caused by the effects of homogenization of microstructure of the material, observed in this range of the plastic strain rate. A clear effect of changes in the mechanical properties of zinc is observed in the range of higher rates of plastic strain, i.e., between 7 s*^−^*^1^ and 170 s*^−^*^1^, which occurred in the process of hydrostatic extrusion. The value of the ultimate tensile strength UTS increases additionally by about 30% and the value of the yield strength YS by about 20%. The strongest effect is observed when comparing the obtained elongation values, which is over seven times higher in this range of strain rate, reaching the maximum value of A = 49% at the strain rate of 170 s*^−^*^1^. The observed differences are the result of the strong homogeneity of the material microstructure, which is the result of continuous dynamic recrystallization taking place during plastic strain at high strain rates. Figure 8 compares the obtained results with selected literature data. Higher elongation values can be obtained only by plastic working carried out at elevated temperatures, as in the case of conventional extrusion processes that were carried out at a temperature of 250 °C [8]. However, due to the high temperature, the possibilities of improving the mechanical properties are limited, which is related to the increase in grain size. After the hot extrusion process, the obtained average grain size was 150 µm. After the HPT (high-pressure torsion) process, the obtained strength value was at the level of UTS = 150 MPa with real strain ε > 5 [25]. It is worth noting that in the present study, zinc was strained to the level of ε ~ 2 and the mechanical properties obtained were much higher, which shows the high efficiency of the HE process. This is also confirmed by the data on mechanical properties obtained by other methods of plastic-working, such as the KOBO process, or ECAP method, which utilizes the Bc method [5,26]. The HE process, in contrast to other methods that generate plastic strains, such as HPT or ECAP, enables plastic processing of large volumes of material with homogeneous mechanical properties, and high hydrostatic pressure and the accompanying high rates of plastic strain ensure the possibility of producing solid products with uniform mechanical and structural properties.

## 5. Conclusions

➢ Thanks to the use of unconventional methods for generating severe plastic deformation (SPD), namely the ECAP and HE processes, the influence of the plastic strain rate on the mechanical and structural properties of pure zinc in a wide range of plastic strains, between 0.04 s^−1^ and 170 s^−1^ was investigated.➢ The test performed showed a strong dependence of changes in the microstructure on the plastic strain rate. For the rates in the range of 0.04 s^−1^ to 0.53 s^−1^ a bimodal microstructure was observed. Increasing the strain rate to 7 s^−1^ caused a significant change in the nature of the microstructure. The microstructure was homogeneous with equiaxed grains free from visible defects inside, which was related to the process of continuous dynamic recrystallization.➢ The maximal plastic strain rate 170 s^−1^ resulted in obtaining the highest mechanical properties (UTS = 194 MPa, YS = 145 MPa) with the highest value of elongation (A = 49%), which in comparison to the material with bimodal microstructure increased more than seven-fold.➢ The results obtained showed that the rate of plastic strain is a critical parameter in the processes generating large plastic deformations. Apart from the plastic deformation rate, which was constant for all the tests performed (ε ~ 2), the plastic strain rate can also have a significant effect on the final properties of the materials. This is of particular importance for materials with low melting points that are susceptible to the heat effects that occur during the plastic-working, such as the tested zinc, which recrystallizes at room temperature.

## Figures and Tables

**Figure 1 materials-15-04892-f001:**
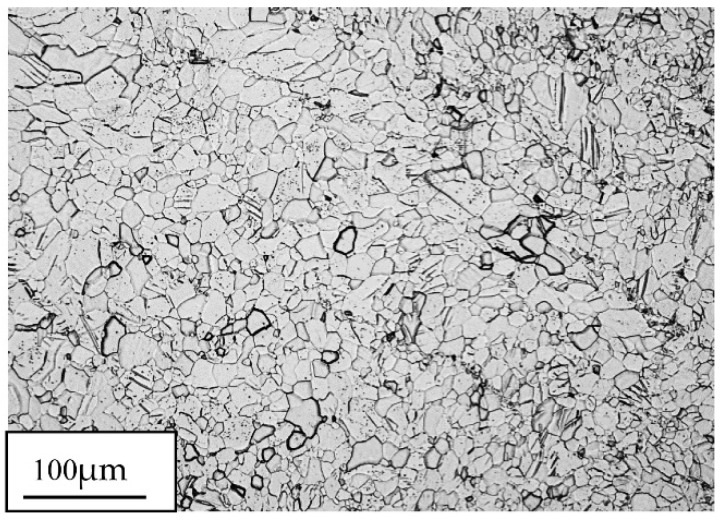
The 99.9% zinc microstructure in the initial state, after a 30 min annealing at 150 °C.

**Figure 2 materials-15-04892-f002:**
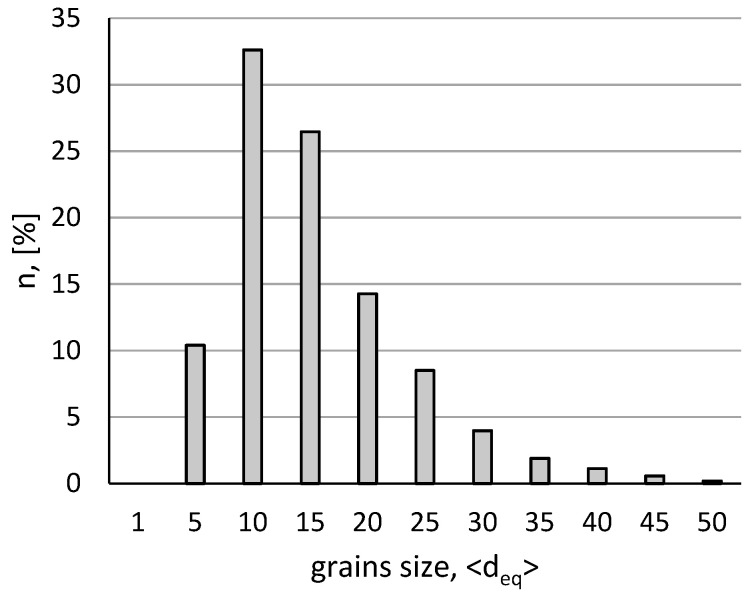
The grain size distribution in zinc was 99.9% in the initial state, after a 30 min annealing at 150 °C.

**Figure 3 materials-15-04892-f003:**
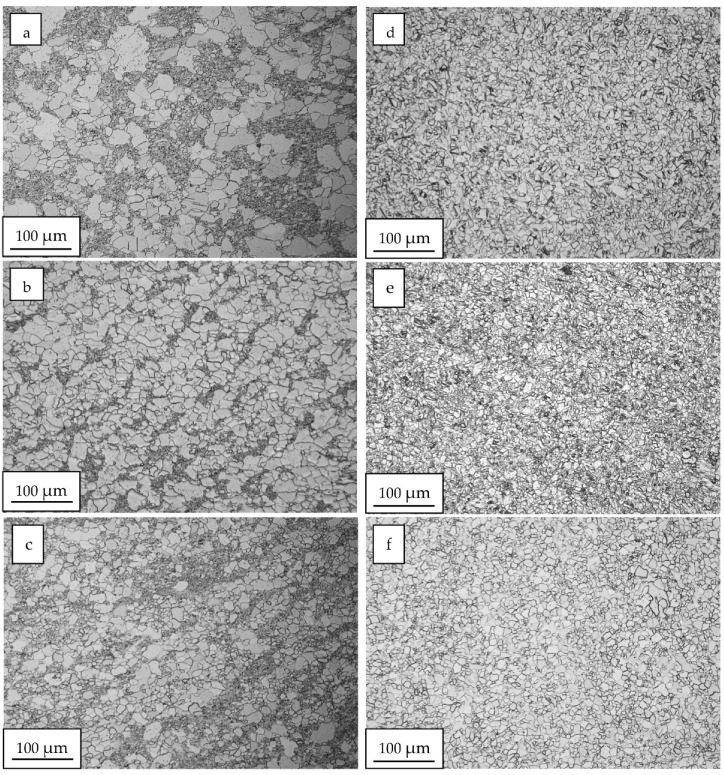
Images of the zinc microstructure after plastic strain using the ECAP and HE processes with the rates of 0.04 s^−1^ (**a**), 0.13 s^−1^ (**b**), 0.53 s^−1^ (**c**), 7 s^−1^ (**d**), 50 s^−1^ (**e**), 170 s^−1^ (**f**).

**Figure 4 materials-15-04892-f004:**
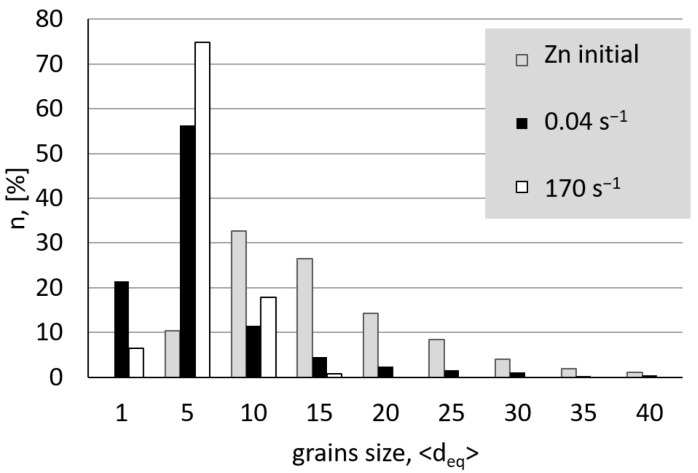
Grain size distribution in the zinc deformed with the lowest strain rate of 0.04 s^−1^, and the highest strain rate of 170 s^−1^ compared to the unprocessed material.

**Figure 5 materials-15-04892-f005:**
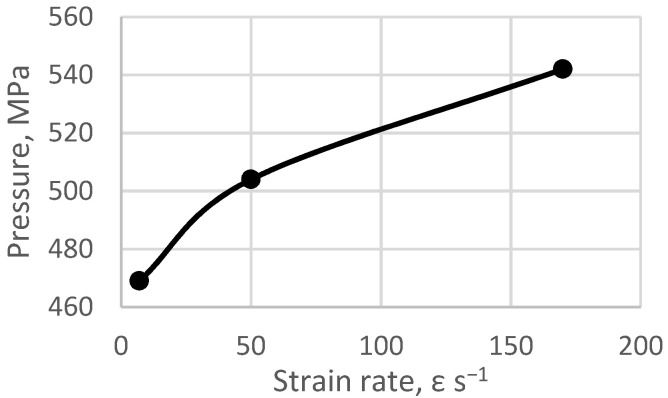
Influence of the strain rate during the HE process on the 99.9% zinc extrusion pressure.

**Figure 6 materials-15-04892-f006:**
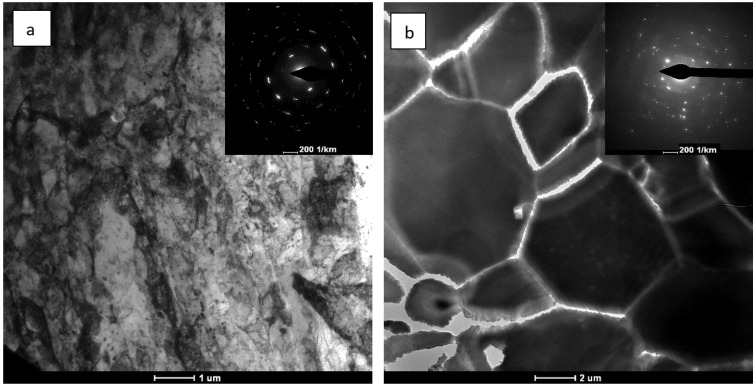
TEM images of the 99.9% zinc microstructure after plastic strain with a strain rate of 0.04 s^−1^ (**a**) and 170 s^−1^ (**b**).

**Figure 7 materials-15-04892-f007:**
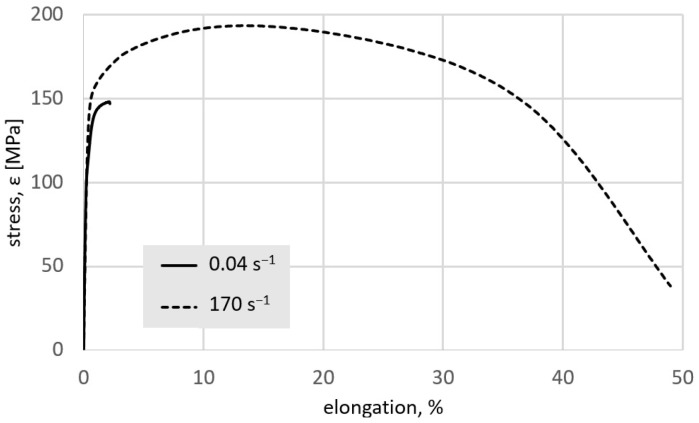
Curves of elongation as a function of stress obtained in the static tensile test of a test piece made of the zinc after plastic strain with the lowest and the highest strain rate (0.04 s^−1^ and 170 s^−1^, respectively).

**Figure 8 materials-15-04892-f008:**
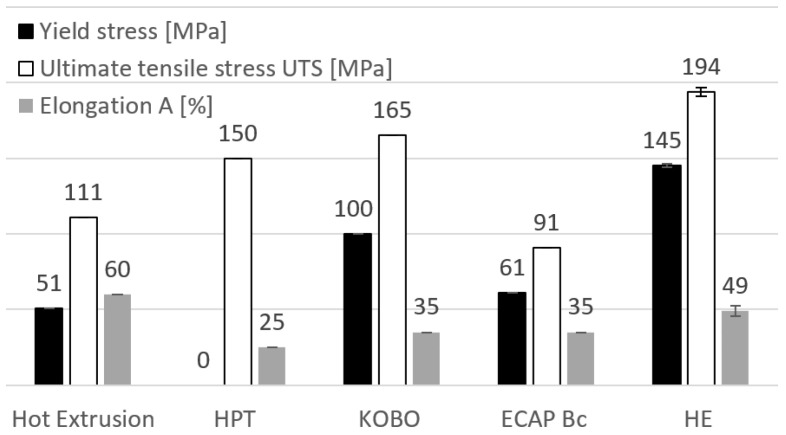
Comparison of the obtained mechanical properties of zinc at the highest strain rate of 170 s*^−^*^1^ with the data in the literature on the subject [5,8,25,26].

**Table 1 materials-15-04892-t001:** The mean grain size d_2_ and the coefficient of variation of the Cv_d2_ distribution for zinc after plastic strain of different rates.

Strain Rate	0	0.04 s^−1^	0.13 s^−1^	0.53 s^−1^	7 s^−1^	50 s^−1^	170 s^−1^
**d_2_ [µm]**	15.3 +/−0.46	6.1 +/−0.36	5.4 +/−0.12	5.9 +/−0.31	7.1 +/−0.19	5.6 +/−0.20	5.5 +/−0.24
**CV_d2_**	0.5	1.05	0.86	0.81	0.36	0.39	0.43

**Table 2 materials-15-04892-t002:** Zinc static rupture test data as a function of strain rate.

Strain Rate	0	0.04 s^−1^	0.13 s^−1^	0.53 s^−1^	7 s^−1^	50 s^−1^	170 s^−1^
**UTS** **[MPa]**	60 +/−6	148 +/−5.2	144 +/−3	148 +/−2.5	190 +/−1.43	192 +/−1.49	194 +/−2.48
**YS [MPa]**	30 +/−4.1	116 +/−1.43	118 +/−3.79	119 +/−2.48	130 +/−2.86	135 +/−1.43	145 +/−4.96
**A [%]**	2 +/−0.8	2 +/−1.3	2.9 +/−1.4	5.9 +/−0.9	42 +/−4.16	47 +/−2.79	49 +/−1.13

## Data Availability

Experimental methods and results are available from the authors.

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
