# Peer review of "Influence of Strain Rates during Severe Plastic Strain Processes on Microstructural and Mechanical Evolution in Pure Zinc"

_materials, 2022, doi:10.3390/ma15144892_

Round 1

Reviewer 1 Report

Dear Authors,

Your work seams new and useful but minor revisions are necessary, as follows.

1. Please do not introduce abbreviations without explaining them. Example: UTS, YS, LM and TEM at page 2 of the PDF file. Please check all manuscript for such problem.

2. Please insert, where available, the apparatus error.

3. Uncertainties in Tables will be useful.

4. Page 3 of the PDF file: “value of the particle size distribution variation coefficient was Cvd2= 0.5”- please insert the method of calculation of the particle size distribution variation coefficient.

5. Page 6 of the PDF file: “The zinc after the straining process with the lowest strain rate 155 of ~ 0.04 s-1 is visible in the areas where defective grains are smaller, Figure 6 a”- this sentence is unclear, please improve it.

6. Figure 8 must be improved since a part of it is covered by the written part.

Reviewer 2 Report

About this article “Influence of strain rates during severe plastic strain processes  on microstructural and mechanical evolution in pure zinc . There is good data for supporting a scientific discussion in the manuscript. I hope readers will enjoy reading this article. This is an interesting study in which, methods and experiments are documented in detail. Results are presented in an appropriate manner, clearly explaining the results of the study. The manuscript is written in a direct and active style. Overall, the paper offers enough details of its methodology to reproduce the experiments and is written comprehensively enough to be understandable.

 I have some minor comments below. The authors are advised to take into consideration the following suggestions:

1) Authors should carefully revise and corrected all the grammatical issues and follow the scientific norms in the whole manuscript

2) Resolutions of Figures can be improving.
Please add FTIR and XRD tests to this paper.

3) Please use updated and recent papers in the literature review to give more sense to the reader.

4 )Conclusions could be more specific and to the point, I would suggest looking and thinking about it.

5) Please more elaborate on the novel aspect of your work at the end of the introduction

Some of introduction on antibacterial and cells applications and … are need using from below papers, so, below papers are add to manuscript Recent publications

1:Bio-Enhanced Polyrhodanine/Graphene Oxide/Fe3O4 Nanocomposite with Kombucha Solvent Supernatant as Ultra-Sensitive Biosensor for Detection of Doxorubicin Hydrochloride in

2: Modification of the epoxy resin mechanical and thermal properties with silicon acrylate and montmorillonite nanoparticles

3: Improved morphology and properties of nanocomposites, linear low density polyethylene, ethylene-co-vinyl acetate and nano clay particles by electron beam

4: Reinforced Polypyrrole with 2D Graphene Flakes Decorated with Interconnected Nickel-Tungsten Metal Oxide Complex Toward Superiorly Stable Supercapacitor

5: Superior X-ray radiation shielding effectiveness of biocompatible polyaniline reinforced with hybrid graphene oxide-iron tungsten nitride flakes
